# *LRTuner*: A Learning Rate Tuner for Deep Neural Networks

**Nikhil Iyer** [*†]    **V Thejas** [*†]
**Nipun Kwatra**    **Ramachandran Ramjee**    **Muthian Sivathanu**

*Microsoft Research India*

IYERNIKHIL007@GMAIL.COM    THEJASVENKATESH97@GMAIL.COM
NIPUN.KWATRA@MICROSOFT.COM    RAMJEE@MICROSOFT.COM    MUTHIAN@MICROSOFT.COM

## Abstract

One very important hyperparameter for training deep neural networks is the learning rate schedule of the optimizer. The choice of learning rate schedule determines the computational cost of getting close to a minima, how close you actually get to the minima, and most importantly the kind of local minima (wide/narrow) attained. The kind of minima attained has a significant impact on the generalization accuracy of the network. Current systems employ hand tuned learning rate schedules, which are painstakingly tuned for each network and dataset. Given that the state space of schedules is huge, finding a satisfactory learning rate schedule can be very time consuming. In this paper, we present *LRTuner*, a method for tuning the learning rate as training proceeds. Our method works with any optimizer, and we demonstrate results on SGD with Momentum, and Adam optimizers.

We extensively evaluate *LRTuner* on multiple datasets, models, and across optimizers. We compare favorably against standard learning rate schedules for the given dataset and models, including ImageNet on Resnet-50, Cifar-10 on Resnet-18, and SQuAD fine-tuning on BERT. For example on ImageNet with Resnet-50, *LRTuner* shows up to 0.2% absolute gains in test accuracy compared to the hand-tuned baseline schedule. Moreover, *LRTuner* can achieve the same accuracy as the baseline schedule in 29% less optimization steps.

## 1. Introduction

Learning rate is one of the most important hyperparameters that impact deep neural network (DNN) training performance. It determines the speed of reaching the minima, impacting the computational cost of optimization, e.g. higher learning rates may get in the neighborhood of a minima faster than lower learning rates (Nar and Sastry, 2018). The learning rate also determines the kind of minima (e.g. wide vs narrow) attained (Keskar et al. (2016); Wu et al. (2018); Jastrzebski et al. (2019); Iyer et al. (2020)), which has a significant impact on the generalization accuracy.

Therefore, it is not surprising that a lot of effort has gone into automatically tuning the learning rate (Schaul et al. (2013); Smith (2017); Rolinek and Martius (2018); Carvalho et al. (2021)). However, these techniques have not been able to deliver state of the art *test accuracy* on standard benchmarks. Instead, deep learning researchers today rely on a mixture of brute force or random search, augmented with simple decay heuristics such as

---

*. Equal contribution

†. Work done during an internship at Microsoft Research India

using a staircase, polynomial, cosine or exponential decay-based learning rate schedules. Jin et al. (2021) show promising results in automatically tuning the learning rate on standard benchmarks, but their method takes over $2\times$ the total training time.

This problem is further exacerbated by the fact that different optimizers such as SGD, Adam work best on different datasets and require very different learning rate schedules. For example, image datasets such as Cifar-10 (Krizhevsky et al., 2009), ImageNet (Russakovsky et al., 2015) perform very well with SGD-momentum optimizer and use a staircase learning rate schedule; while NLP tasks such as machine translation (Vaswani et al., 2017) and language modeling (Devlin et al., 2018) typically use Adam optimizer with a learning rate schedule consisting of a linear warmup followed by polynomial decay.

Our learning rate tuning scheme is based on taking a quadratic approximation of the local loss landscape (Section 2). We first probe for loss values attained by the optimizer, as the current learning rate is perturbed by small values. Since we only make small perturbations, the loss function can be modeled *locally* via a Taylor series expansion. We then make a quadratic approximation and solve for the optimal learning rate perturbation which minimizes the loss. This is similar to Newton's method, but applied only in the descent direction. Extensive empirical analysis validates that this quadratic approximation works well even for large DNNs. Although the basic idea is simple, it is complicated by the use of minibatch based stochastic optimization methods in DNNs. The loss value computed from a minibatch is a noisy estimate of the true loss, and can throw off our approximation. We thus propose a few techniques to handle stochasticity.

Our method for learning rate tuning is a local method, and does not take into account the non-convex global optimization landscape of deep networks. As a result, it can take locally optimal, but globally suboptimal decisions which can impact generalization. To tackle this, we take inspiration from Iyer et al. (2020), which proposes an *Explore-Exploit* based learning rate schedule to allow the optimization to land in a wide minimum by training at a high learning rate, before starting descent into this minimum by using a linear decay schedule. We found that incorporating such an *Explore* phase into our method significantly helped with generalization.

We demonstrate the efficacy of *LRTuner* across a wide range of models and datasets, including SQuAD on BERT-base, Transformer on IWSLT'14 (DE-EN), ResNet-50 on ImageNet, Cifar-10 on ResNet18; and across multiple optimizers: SGD-Momentum and Adam. In all cases, compared to the original hand-tuned learning rate baselines, *LRTuner* matched or exceeded the test accuracy when trained using the original training budget. We also achieve reduction of 29% and 24% training steps on ImageNet and IWSLT'14(DE-EN) while matching baseline accuracies, resulting in significant gains in wall clock training time.

## 2. Method

Let $L(\theta)$ be the loss of the network as a function of its parameters $\theta$. Note that in practice, since we work with stochastic optimizers such as SGD, the loss computed in each minibatch is only an approximation of the true loss. This distinction is important when we come to the estimation of the best learning rate, but can be ignored for the formulation below.

The loss of the network in the next time step is $L(\theta - \eta\vec{d})$, where $\vec{d}$ is the step direction, and $\eta$ is the learning rate. If we think of $L(\theta - \eta\vec{d})$ as a function of the learning rate $\eta$, our task is to find an $\eta$ which minimizes this function. Since it is hard to directly estimate $L$

as a function of $\eta$, we instead consider what happens if we perturb $\eta$ by a small amount $\epsilon$, i.e we look at $L(\theta - (\eta + \epsilon)\vec{d})$. Looking at this as a function of $\epsilon$, and applying Taylor series expansion we get,

$$\hat{L}(\epsilon) = L(\theta - (\eta + \epsilon)\vec{d}) = L(\theta - \eta\vec{d} - \epsilon\vec{d})$$
$$= L(\theta - \eta\vec{d}) - \epsilon\vec{d}^T\vec{g} + \frac{1}{2}\epsilon^2\vec{d}^T H\vec{d} + \mathcal{O}(\epsilon^3), \tag{1}$$

where $\vec{g}$ is the gradient of $L$ w.r.t $\theta$, and $H$ is the Hessian. Note that computing accurate value of the gradient $\vec{g}$ is expensive, as it requires going over the entire dataset (not just a minibatch), while computing the Hessian $H$ is typically intractable for deep networks which have millions of parameters. However, it turns out that we don't actually need the values of $\vec{g}$ and $H$. In fact, we can just look at $\hat{L}$ as a quadratic function of $\epsilon$, i.e.

$$\hat{L}(\epsilon) = k_0 + k_1\epsilon + k_2\epsilon^2 + \mathcal{O}(\epsilon^3). \tag{2}$$

To compute $\{k_0, k_1, k_2\}$, we simply evaluate the loss at a few values of $\epsilon$ and fit a quadratic polynomial. This can be done at the cost of one forward pass per $\epsilon$ sample. Note that we are able to get away with having to calculate the high dimensional $\vec{g}$, and $H$, because the loss function only looks at the effect of these first and second order derivatives in *one single direction* $\vec{d}$, as determined by the terms $\vec{d}^T\vec{g}$ and $\vec{d}^T H\vec{d}$. Once we know the values of $\{k_0, k_1, k_2\}$, we can simply minimize this quadratic to get the optimal perturbation on the current learning rate ($\epsilon_{min} = -\frac{k_1}{2k_2}$), to minimize the loss $L$ in the next time step. Observe that our method only needs the search direction $\vec{d}$ for computing the various loss samples. As a result, our method works for any optimizer by simply using its search direction in the above formulation. To reduce computational cost, we tune the learning rate only every few minibatches (See Appendix D for details on computation cost of our method).

We perform extensive empirical analysis in Appendix A to validate that this quadratic approximation works well even for large DNNs. Note that it is very important to use a perturbation $\epsilon$ for the Taylor expansion in Equation 1, rather than just expanding on $\eta$ via $L(\theta - \eta\vec{d}) = L(\theta) - \eta\vec{d}^T\vec{g} + \frac{1}{2}\eta^2\vec{d}^T H\vec{d} + \mathcal{O}(\eta^3)$. This is because $\eta$ may not be small, causing the $\mathcal{O}(\eta^3)$ error of the approximation to be quite large.

**Epsilon Thresholding.** Note that the quadratic approximation in Equation 2 is valid only up to an $\mathcal{O}(\epsilon^3)$ error. However, the optimal learning rate perturbation computed as the quadratic minimum ($\epsilon_{min} = -\frac{k_1}{2k_2}$), can be quite large at times, causing the approximation error to blow up. This is illustrated in Figure 3 (Appendix). To avoid this, we threshold the estimated $\epsilon_{min}$ to a small value. Since we are interested in approximating the loss to some precision, we use a relative threshold as follows. We approximate $\mathcal{O}(\epsilon^3)$ as $|\epsilon|^3$, and bound the error:

$$|\epsilon|^3 < r * L(\theta), \tag{3}$$

where $r$ is the epsilon threshold initialized before training (Refer Appendix E for details).

**Stochasticity:** For computation of the quadratic coefficients in Equation 2, we need to compute the loss values $\hat{L}(\epsilon)$ at multiple values of $\epsilon$. Note that in a typical DNN setting

we never compute the full loss, but only a stochastic loss based on a given minibatch. This loss, however, can be noisy and throw off our estimate of $\epsilon_{min}$. To handle this we follow two simple strategies. First, we use a bigger minibatch (called *superbatch*) for computing the loss, and second, we use the *same* superbatch for computing all losses for a particular estimate. We use an integer multiple of minibatches to make a superbatch. This allows us to compute the superbatch loss by simply averaging multiple single batch forward passes, and does not increase the peak GPU memory usage. See Appendix C for details on selection of superbatch size.

Finally, as a safeguard we also added a *rollback* policy. In case our system makes a bad call on the learning rate change (because of a bad superbatch sample), which leads to a reduction in loss drop per iteration, we rollback the decision and revert the state of the network to the time we made the learning rate change. Although, *rollback* triggers rarely, it is helpful in preventing the optimization from going astray because of one bad decision.

## 2.1 Generalization

Due to the *local* nature of our method, it can take locally optimal, but globally suboptimal decisions. This is a problem for DNNs, which have highly non-convex loss landscapes, and such a local method can lead to bad generalization. To tackle this, we take inspiration from the hypothesis proposed in Iyer et al. (2020). They hypothesize that the density of wide minima is far lower than narrow minima, and emphasize on the need to train at a high learning rate for sufficient period, even if the training loss stagnates. The high learning rate training helps the optimizer escape narrow minima and land in a wide minimum with high probability. Since wide minima are shown to generalize better than narrow minima (Keskar et al., 2016; Jastrzebski et al., 2017; Wang et al., 2018; Chaudhari et al., 2019), this leads to better generalization. They propose an *Explore-Exploit* LR schedule where the explore phase scans the landscape with a high LR for sufficient duration to land in a wide minimum, followed by an exploit phase where they descend into this minimum.

Since *LRTuner* greedily finds the optimal learning rate for maximal reduction in the local training loss, it may reduce the learning rate too quickly and this often generalizes badly. To circumvent this, we add an *Explore* phase as recommended in Iyer et al. (2020), where we train the network at only high learning rates. During this phase, *LRTuner* only allows increases in learning rate proposals from the local quadratic approximation of Equation 2. Any proposals to decrease the learning rate are rejected. This is then followed by an *Exploit* phase, where *LRTuner* permits proposals to reduce the learning rate and does not allow increases. This two phase scheme achieves good generalization performance.

**Recent work on generalization:** Although understanding generalization of deep neural networks is an open problem, there have been interesting findings recently. Recent works show that wide minimas generalize much better than narrow minimas (Hochreiter and Schmidhuber (1997); Arora et al. (2018); Keskar et al. (2016); Jastrzebski et al. (2017); Wang et al. (2018)), even though they have the same training loss. Chaudhari et al. (2019) design entropy based functions to drive optimizers into flatter surfaces. Iyer et al. (2020); Wu et al. (2018); Baldassi et al. (2019) show that the density of wide minima are far lower than narrow minima. Kawaguchi (2016) state that neural landscapes have multiple local minimas, but all local minima are also the global minima (also see Goodfellow et al.

(2016)). Keskar et al. (2016) found that small batch SGD generalize better than large batch SGD and also lands in wider minimas, suggesting that noise in SGD acts as an implicit regularizer. Hoffer et al. (2017) propose to close the generalization gap in large batch training by increasing the number of optimization steps to match small batch regime. Jastrzebski et al. (2017) suggest that ratio of learning rate to batch size plays a key role in determining the final minima width and a larger ratio leads to better generalization. More recent work have been able to generalize quite well even with very large batch sizes. Goyal et al. (2017); McCandlish et al. (2018) scale the learning rate linearly as a function of batch size, while You et al. (2020) pre-train BERT$_{\text{Large}}$ in 76 mins with square-root LR scaling.

## 3. Experiments

We extensively evaluate our method on multiple networks and datasets, as well as multiple optimizers including SGD, Momentum and Adam. We have implemented *LRTuner* as an optimizer in PyTorch (Paszke et al. (2017)), which wraps any existing optimizer. For our experiments, we employ an out of the box policy as Rolinek and Martius (2018), where we just wrap the existing optimizer with *LRTuner*, and do not modify anything else. We evaluate on multiple image datasets – Imagenet on Resnet-50, Cifar-10 on Resnet-18; as well as NLP datasets – Squad v1.1 for BERT finetuning and IWSLT'14(DE-EN) on Transformers.

### 3.1 ImageNet image classification on Resnet-50

In this experiment we trained the ImageNet dataset (Russakovsky et al. (2015)) on Resnet-50 (He et al., 2016) network [1]. We evaluated our method on SGD with momentum of 0.9, weight decay of $1e^{-4}$ and a batch size of 256, similar to popular baselines. The baseline runs use a standard step schedule of of 0.1, 0.01 and 0.001 learning rate for 30 epochs each. With *LRTuner*, we trained the network with 25 explore epochs, and used the same seed learning rate as the baseline schedule, i.e. 0.1. Table 1 shows the training loss and test accuracies for the various runs. As shown, *LRTuner* improves on the test accuracy of the baseline by a comfortable margin. We also observed that *LRTuner* can reach the baseline top-1 accuracy of 75.87 around 64 epochs (29% reduction) in all our runs. See Figure 6 for comparisons of training loss, test accuracy, and learning rate.

| LR Schedule | Training Loss | Test Top 1 Acc. | Test Top 5 Acc. |
|:---:|:---:|:---:|:---:|
| Baseline | 0.74 (0.001) | 75.87 (0.035) | 92.90 (0.015) |
| *LRTuner* | 0.74 (0.041) | 76.06 (0.098) | 93.03 (0.024) |

Table 1: Training loss and Test accuracy for ImageNet on Resnet-50. We report the mean and standard deviation over 3 runs.

### 3.2 Cifar-10 image classification on Resnet-18

In this experiment we trained the Cifar-10 dataset (Krizhevsky et al. (2009)) on Resnet-18 network (He et al. (2016)) [2]. We evaluated our method on SGD with momentum of 0.9,

---

1. We used the implementation at: https://github.com/cybertronai/imagenet18_old
2. We used the implementation at: https://github.com/kuangliu/pytorch-cifar

weight decay of $5e^{-4}$ and batch size 128. For baseline runs, we used a standard step schedule of 0.1, 0.01 and 0.001 learning rate for 100, 50, 50 epochs; while for *LRTuner*, we trained the network with 100 explore epochs, and used the same seed learning rate as baseline, i.e. 0.1. Table 2 shows the training loss and test accuracy for the various runs. As shown, *LRTuner* achieves nearly the same test accuracy as baseline. See Figure 5 for more detailed comparisons of training loss, test accuracy, and learning rate.

| LR Schedule | Training Loss | Test Acc |
|---|---|---|
| Baseline | 0.002 ($6.5e^{-5}$) | 94.81 (0.001) ) |
| *LRTuner* | 0.0008 ($1e^{-4}$) | 94.79 (0.001) |

Table 2: Training loss and Test accuracy for Cifar-10 on Resnet-18. We report the mean and standard deviation over 7 runs.

### 3.3 SQuAD fine-tuning on BERT

We now evaluate *LRTuner* on a few NLP tasks. In the first task, we fine-tune the BERT$_{\text{BASE}}$ model (Devlin et al. (2018)) on SQuAD v1.1 (Rajpurkar et al. (2016)) with the AdamW optimizer[3]. Fine-tuning is typically run for only a few epochs. We use the standard baseline which trains for 2 epochs with a seed learning rate of $2e^{-5}$ with linear decay. The *LRTuner* runs were trained with 2500 explore steps ($\approx$ half epoch), and the same seed learning rate of $2e^{-5}$ as baseline. Table 3 shows our results over 3 runs. We achieve an EM score of 81.2, compared to baseline's of 80.7. Moreover, we found that *LRTuner* can reach the baseline accuracy of 80.7 in 20% less training steps. See Figure 8 for detailed comparisons.

| LR Schedule | Train Loss (av) | EM (av) | F1 (av) |
|---|---|---|---|
| Baseline | 0.96 (0.075) | 80.7(0.18) | 88.2 (0.02) |
| *LRTuner* | 1.05 (0.008) | 81.2 ((0.52) | 88.5 (0.09) |

Table 3: SQuAD fine-tuning on BERT. We report the average training loss, average test EM and F1 scores over 3 runs.

### 3.4 Machine Translation on Transformer Network with IWSLT

In the second NLP task, we train the Transformer network (Vaswani et al. (2017)) on the IWSLT German-to-English (De-En) dataset (Cettolo et al. (2014)) with the Adam optimizer [4]. For baseline, we used the learning rate schedule mentioned in Vaswani et al. (2017). The baseline learning rate starts at $1.25e^{-7}$, and is linearly increased for 4000 steps to $5e^{-4}$, followed by an inverse square root decay till 50 epochs. The training batches consist of approximately 4000 tokens. With *LRTuner*, we trained the network with 10 explore epochs, and used a seed learning rate of $1e^{-5}$. In both cases we use the model checkpoint with least loss on the validation set for computing BLEU scores on the test set. Table 4 shows the training loss and test accuracy averaged over 3 runs. As shown, *LRTuner* achieves a mean test BLEU score of 34.88, compared to 34.70 for the baseline. Moreover, we observed that

---

3. We used the implementation at: https://github.com/huggingface/transformers
4. We used the implementation at: https://github.com/pytorch/fairseq

*LRTuner* can reach the baseline BLEU score of 34.70 around 38 epochs (24% reduction) in all our runs. See Figure 7 for detailed comparisons of training/validation perplexity, learning rate, etc.

| LR Schedule | Train ppl | Validation ppl | Test BLEU Score |
|---|---|---|---|
| Baseline | 3.55 (0.029) | 5.10 (0.033) | 34.70 (0.001) |
| *LRTuner* | 3.46 (0.16) | 4.86 (0.014) | 34.88 (0.005) |

Table 4: Training, validation perplexity and test BLEU scores for IWSLT on Transformer networks. The test BLEU scores are computed on the checkpoint with the best validation perplexity. We report the mean and standard deviation over 3 runs.

## 4. Conclusion and Future work

We present *LRTuner*, a novel learning rate tuning method via a local quadratic approximation of the loss landscape in the search direction. We extensively validate *LRTuner* on both image (ImageNet, Cifar-10) and NLP (IWSLT, Squad) datasets, as well as multiple optimizers, and achieve or exceed the test accuracy of original hand tuned learning rate schedules. We also showed that *LRTuner*, in many cases, can achieve the same baseline accuracy in significantly reduced training time. *LRTuner* has a few hyperparameters such as epsilon threshold, and explore epochs. In future work, we would like to completely eliminate these hyperparameters to have a fully automated learning rate tuning scheme.

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

## Appendix A. Validation of Quadratic Approximation

We want to validate whether our quadratic approximation is a good approximation for modelling the loss as a function of the perturbation $\epsilon$. Figure 1 shows a few examples

demonstrating the effectiveness of second order approximation. As shown, the loss values at various samples overlap with the quadratic curve almost perfectly, including those not used in estimation of the quadratic (green triangles). Also, the estimated loss value at the quadratic minima matches the true loss value there quite well. Figure 2 shows quadratic plots for more datasets/models. Figure 3 shows why limiting epsilon range is important.

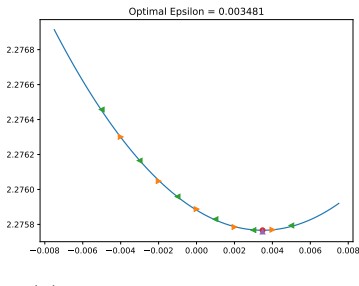

(a) Minimum within range

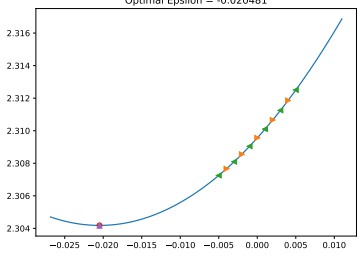

(b) Minimum outside range

Figure 1: Quadratic approximation of loss as a function of $\epsilon$. Shown are two examples from Cifar-10 on Resnet-18 runs where the minimum is (a) within and (b) outside the range of loss samples. The orange triangles show loss samples used in fitting the quadratic, the blue line shows our quadratic approximation, and the green triangles show more loss samples which were not used for fitting. The red circle shows the minimum loss value as per the quadratic, while the purple triangle show the true value at that $\epsilon$.

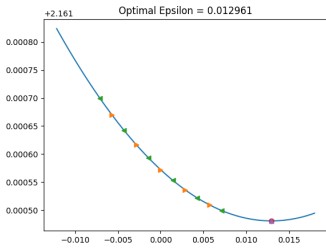

(a) ImageNet on Resnet-50

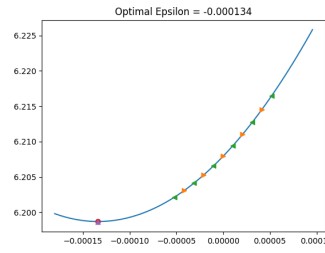

(b) IWSLT on Transformer

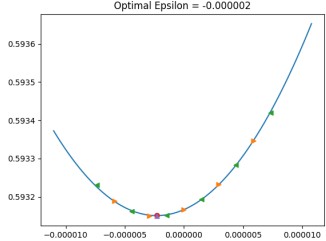

(c) SQuAD fine-tuning BERT

Figure 2: Quadratic approximation curve samples from (a) ImageNet on Resnet-50, (b) IWSLT on Transformer and (c) SQuAD fine-tuning on BERT. The legend is same as figure 1. It can be seen that the quadratic approximation works pretty well.

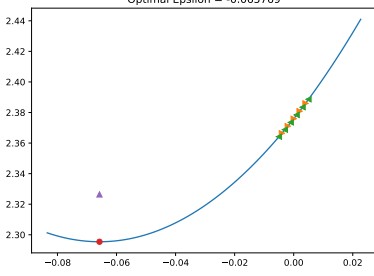

Figure 3: Quadratic approximation error. Orange and green triangles show loss samples used in fitting and testing, respectively. Red circle shows the minimum loss value, and purple triangle shows the true loss at minimum. We clip $\epsilon_{min}$ as mentioned in Equation 3 if $\epsilon_{min}$ exceeds the upper bound.

## Appendix B. Saturation Threshold

*LRTuner* may propose a drop in learning rate during the exploit phase based on the local approximation, which may be sub-optimal globally and could drive the optimizer into local minima or saddle points. To circumvent this, we introduce a *saturation threshold*. The idea is that we want to continue with the current learning rate and not lower it, unless it has *saturated* in terms of loss drop per iteration. That is, if the training loss drop rate has dropped below a threshold for the current learning rate, it suggests that the current learning rate has served its purpose, and we can move on to a lower learning rate if suggested so by *LRTuner*. We can either use an absolute threshold, or a relative threshold where we use the ratio of loss-drop rate when we started using the current learning rate to the current loss-drop rate. Finding the best absolute threshold for each model/dataset can be tricky, so we use a relative threshold, which we found easy to tune. Also, we noticed that the loss drop rate doesn't change drastically towards the end of the training when the loss has anyways stabilized. This can cause a high relative saturation threshold to be too strict in the later stages of training, while it would perform well early on. To handle this, we used a simple strategy where we choose a relative saturation threshold initially, and first time the saturation threshold is crossed, we switch to an absolute threshold with the current loss drop date as the absolute saturation threshold. This essentially amounts to using the relative threshold to determine a good absolute threshold value for the current model and dataset, which is then used subsequently. Refer to Appendix E for more details.

Note that, although both *explore phase* and *saturation threshold* prefer higher learning rates, they serve different purposes. The explore phase allows for only increase in learning rate and ensures that the optimization escapes sharp minima and reaches the neighborhood of a wide minima with a good probability. Saturation threshold is activated only during exploit phase where we strictly reduce the learning rate and do not allow increases. Saturation threshold adds an overall preference for a higher learning rate until that learning rate has served its purpose and is not optimizing the loss satisfactorily.

## Appendix C. Superbatch size selection

To choose an appropriate superbatch size, we measure the standard deviation of loss as a function of superbatch size. We compute this by evaluating the loss with 10 different randomly sampled superbatches and calculate the standard deviation. Figure 4 shows the measurements. We used a conservative superbatch size of 100 in all our examples, as it corresponded to low variance. Note that higher superbatch sizes add an increased computational overhead, but since we recompute our learning rate infrequently the total overhead is not very high.

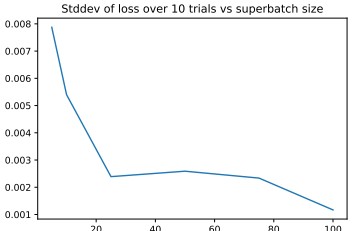

Figure 4: Standard deviation of loss as a function of superbatch size for Cifar10 on Resnet-18

## Appendix D. Computational Cost

The primary computational cost of our method comes from computing the samples for quadratic approximation. We use a superbatch size of 100 and pick 5 samples for ap-

proximating our quadratic, incurring a cost of 500 forward passes each time we need to recompute the learning rate. In most of our examples we keep the *recompute_window* such that our learning rate recomputation occurs once or twice every epoch, except in BERT fine-tuning where we only have 2 epochs to train, and thus want more frequent tuning of the learning rate. In ImageNet on ResNet50 for example, each epoch consists of 5000 steps and we compute the approximation twice every epoch. A backward pass is typically 2x more computationally expensive than forward pass, and 5000 minibatch steps cost 15000 forward pass worth of compute. Our recomputation per epoch costs 1000 forward passes of compute, and thus the computational overhead is $1000/15000, \approx 6.66\%$. Note that in *exploit* phase, we do a recomputation of learning rate only when saturation threshold is crossed. So the computational cost comes down even further and 6.66% overhead is an upper bound. We also achieve baseline accuracy in 29% less training steps, demonstrating significant wall clock time savings compared to baseline schedules. Similarly for IWSLT, we compute our approximation once every epoch (1100 steps). Thus, the upper bound in computational overhead is around $500/3300, \approx 15\%$. We show that *LRTuner* achieves the baseline BLEU score in 24% less training steps , thus showing speedup in training. The automatic tuning method by Jin et al. (2021) also show reductions in training iterations. However, their method takes $2\times$ the total wall clock time as baseline because of high computational overhead.

Note that since the main focus of this work was to develop an automatic learning rate tuning scheme which generalizes as well as state of the art hand tuned learning rate schedules, we have not invested much effort on reducing the computational cost. For example, a 100 superbatch size is highly conservative and a size of 50 or fewer may be sufficient. Similarly, 3 samples for estimating the quadratic are enough most of the times instead of the 5, as the quadratic fit is usually accurate. Also, we can experiment with higher *recompute_window* sizes towards the end of training as mentioned in Jin et al. (2021) and reduce the overhead further.

## Appendix E. Hyperparameters

Table 5 shows the hyperparameters used for all our examples. As shown, the main parameter to tune is the explore epochs, which we typically set to around $20-30\%$ of the total budget. The seed learning rates are same, except for IWSLT, where we had to choose a bigger seed learning rate ($1e^{-5}$), because the baseline seed lr ($1.25e^{-7}$) hindered *LRTuner*'s ability to compute larger epsilons to speed up optimization. On the other hand, *LRTuner* substitutes warmup with explore, which can be explored further theoretically. Selection of seed learning rates can be done with methods proposed in Smith (2017).

The reason for different scale of epsilon thresholds in NLP datasets compared to image datasets, is because they typically run on much lower learning rates (of the order of $1e^{-5}$) compared to image datasets (of the order of $1e^{-1}$ to $1e^{-3}$), which suggests that the optimization landscapes are more sensitive to smaller perturbations and thus need more aggressive clipping. Since $\epsilon \propto$ (epsilon threshold)$^{1/3}$ (Refer to equation 3), we found this hyperparameter easy to set by simply choosing the value to be around twice or thrice the order of the peak learning rate.

We also noticed that the loss drop rate changes much more drastically in image datasets than NLP datasets, again most likely because of the very low learning rates used in NLP datasets. Thus we need to use a lower saturation threshold in NLP experiments compared to image ones. We found this hyperparameter very easy to set, by simply eyeballing at the loss drop rate changes of a trial run with fixed LR for two epochs. This hyperparameter switches to the loss drop rate seen during the first time it is crossed and stays the same for the entirety of exploit phase, as mentioned in Appendix B.

We are also looking at ways to automate the explore epochs hyperparameter, by looking at second order information about the loss landscape during training, and determine if we have reached a wider minima region. See Jastrzebski et al. (2019); Yao et al. (2020) for some interesting analysis on this front.

| Experiments | Explore Epochs | Total Epochs | Seed LR | Saturation Threshold | Epsilon Threshold |
|---|---|---|---|---|---|
| Cifar-10 | 100 | 200 | 0.1 | 100 | $1e^{-3}$ |
| Imagenet | 25 | 90 | 0.1 | 500 | $1e^{-3}$ |
| IWSLT'14 (De-En) | 10 | 50 | $1e^{-5}$ | 5 | $1e^{-8}$ |
| SQuADv1.1 (Bert-Base) | 0.5 | 2 | $2e^{-5}$ | 2 | $1e^{-12}$ |

Table 5: Hyperparameters used for all experiments.

## Appendix F. Detailed Plots

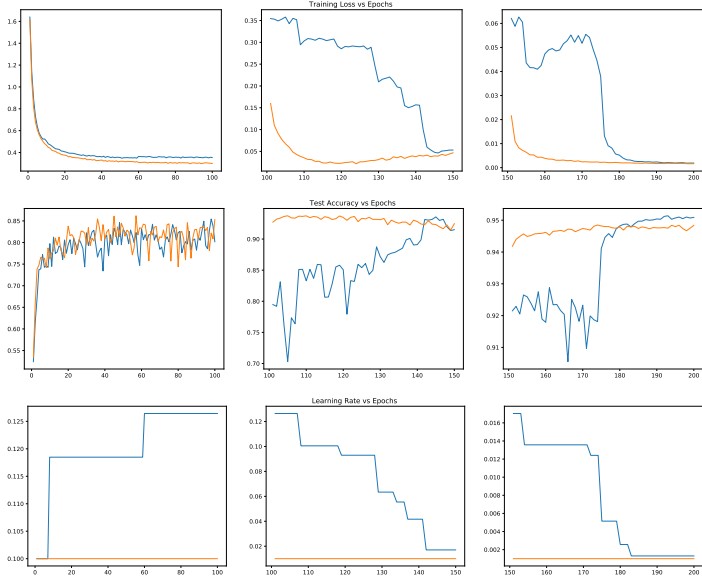

Figure 5: Cifar-10 on Resnet-18 trained with Momentum. Shown are the training loss, test accuracy and learning rate as a function of epochs, for the baseline scheme (orange) vs the *LRTuner* scheme (blue). The plot is split into 3 parts to permit higher fidelity in the y-axis

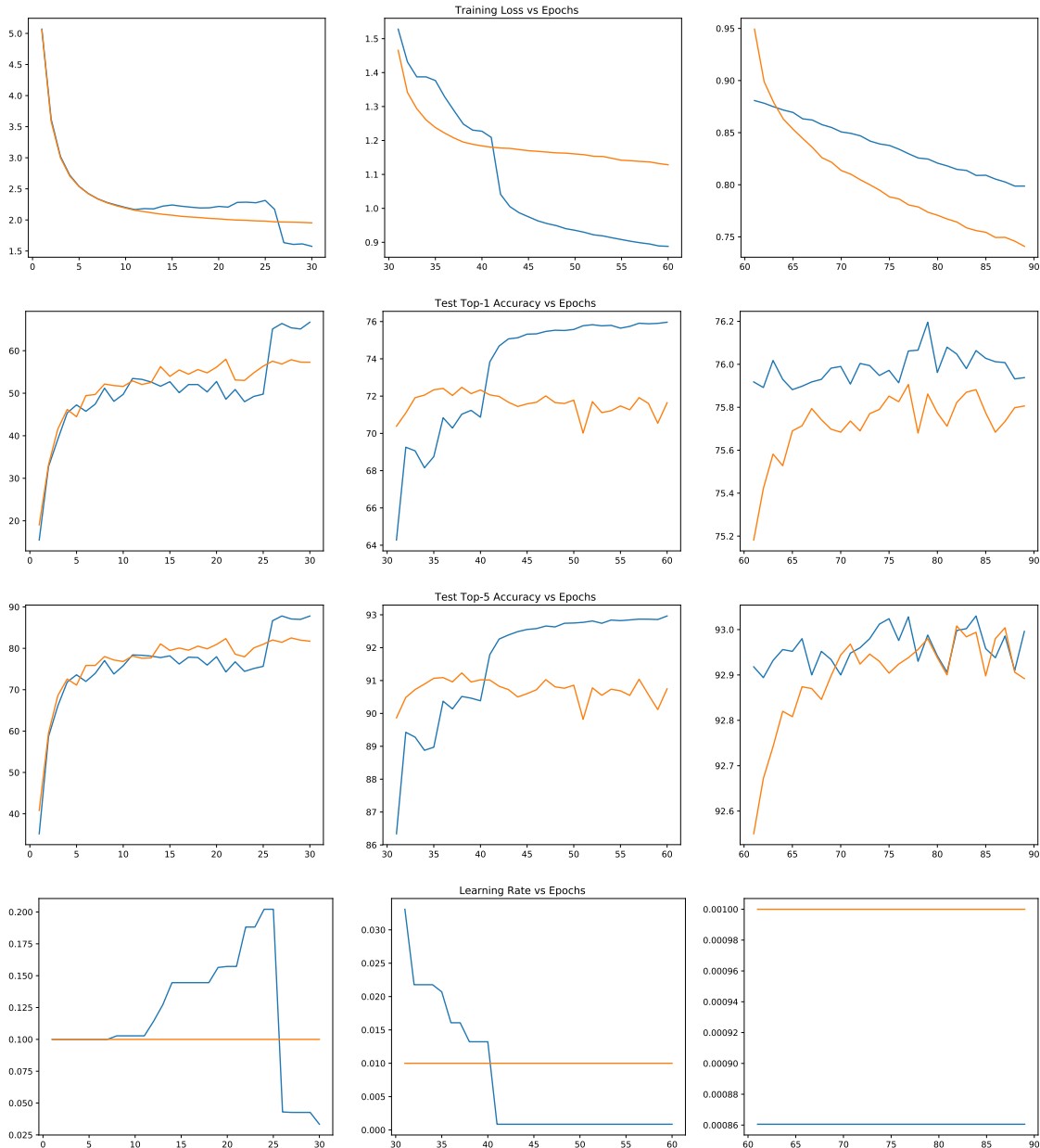

Figure 6: ImageNet on Resnet-50 trained with Momentum. Shown are the training loss, top-1/top-5 test accuracy and learning rate as a function of epochs, for the baseline scheme (orange) vs the *LRTuner* scheme (blue). The plot is split into 3 parts to permit higher fidelity in the y-axis range.

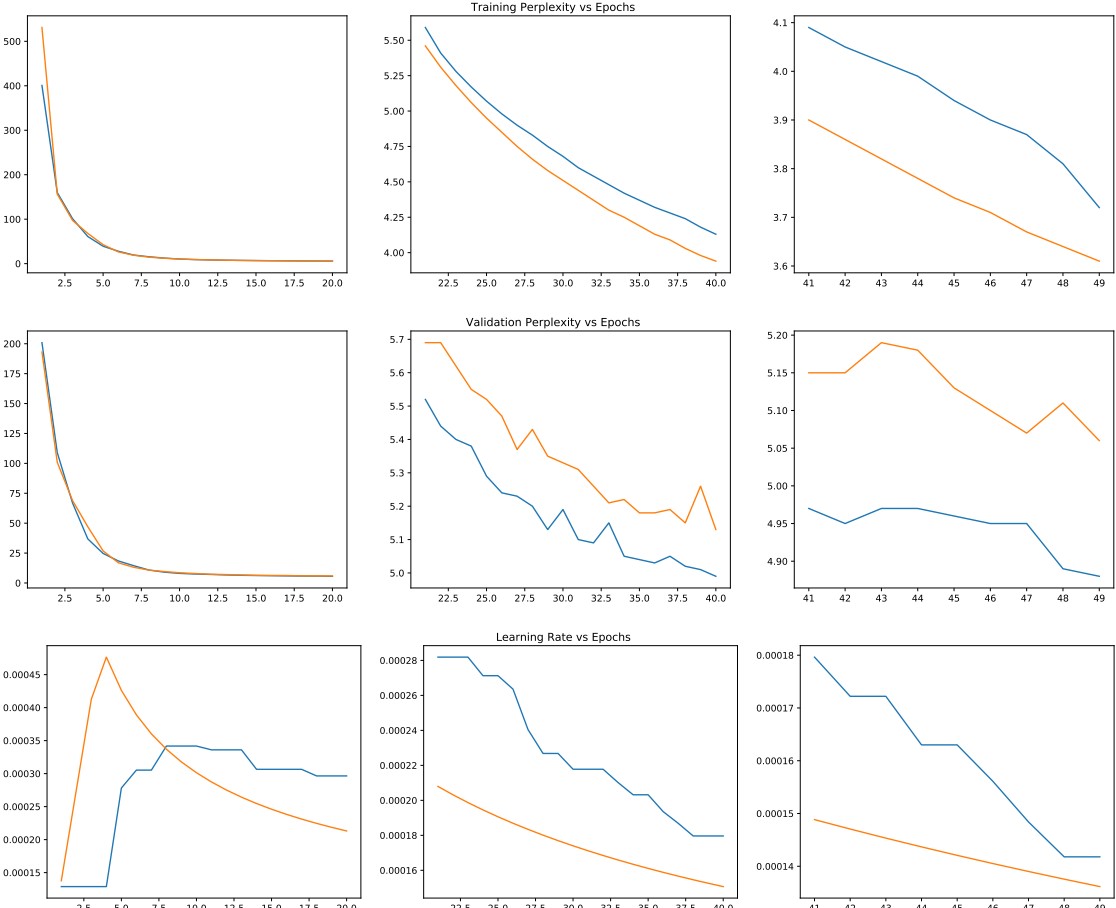

Figure 7: IWSLT on Transformer network trained with Adam. Shown are the training perplexity, validation perplexity and learning rate as a function of epochs, for the baseline scheme (orange) vs the *LRTuner* scheme (blue). The plot is split into 3 parts to permit higher fidelity in the y-axis range.

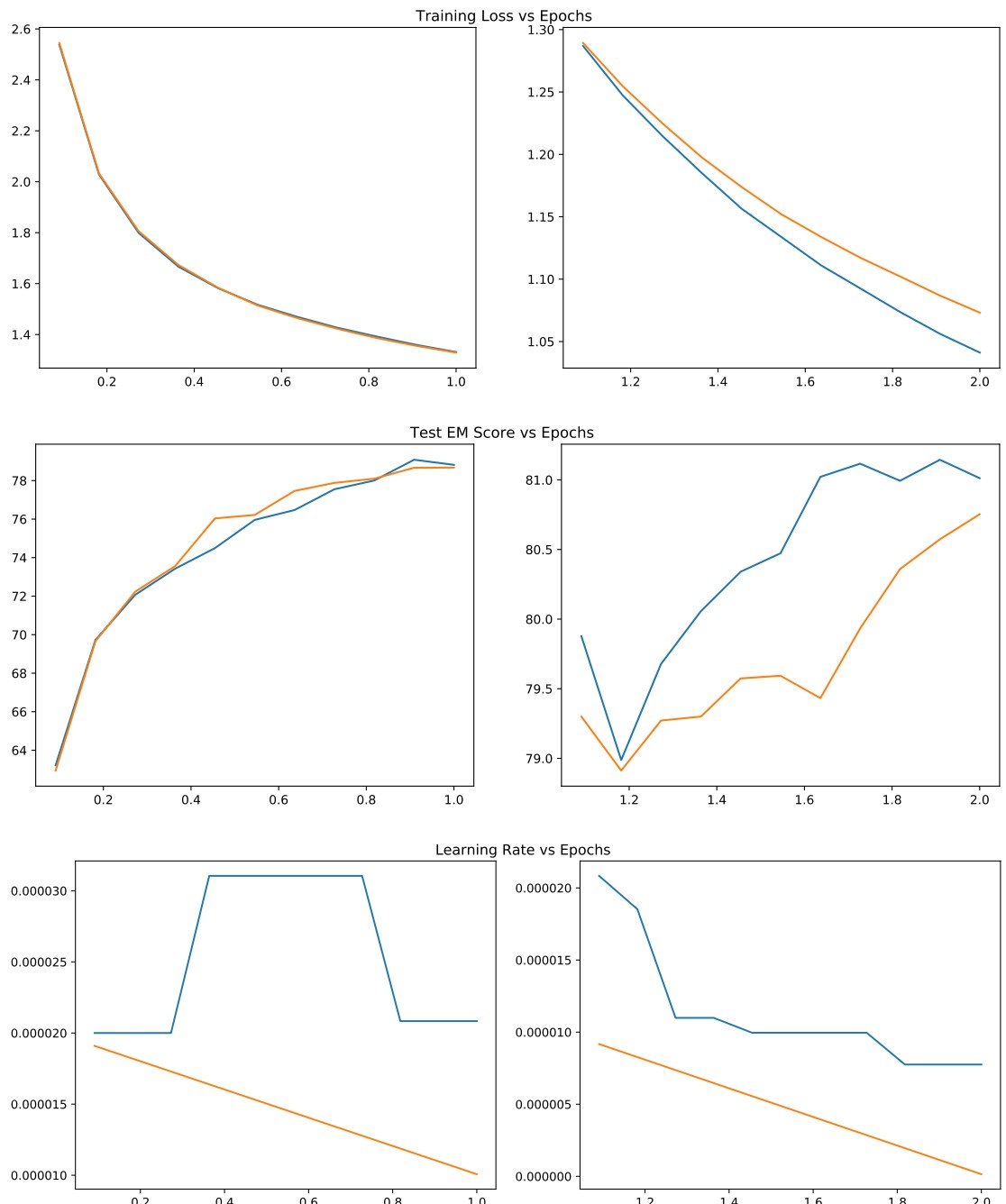

Figure 8: SQuAD fine-tuning on BERT trained with AdamW. Shown are the training loss, test EM score, and learning rate as a function of epochs, for the baseline scheme (orange) vs the *LRTuner* scheme (blue). The plot is split into 2 parts to permit higher fidelity in the y-axis range.

