# OpenReview forum: "LRTuner: A Learning Rate Tuner for Deep Neural Networks"
_ICML.cc/2021/Workshop/AutoML — AutoML@ICML2021 Poster_

### Official Review · Reviewer_5G1e · 2021-06-11
**LRTuner**

**Rating:** 8
**Confidence:** 2

**Review:**

The paper introduces LRTuner, a method by which to automatically tune the learning rate for training deep neural networks.
By evaluating the effect of a few marginally different learning rates they can fit the local loss function and deduce an approximate optimal learning rate for that step (with certain caveats).
The method is agnostic to the optimizer used, since it only relies on the step direction, but does introduce some minor overhead (of a few 'superbatch' forward passes) which is negligible.

From the introduction to Section 3, I assume default values for episilon threshold were used in the evaluation of LRTuner.
I would advise to state this explicitly (and if it's not the case, state which values were used). In either case it should probably be reported on which tasks the (default) value of ` the epsilon threshold were tuned (if it was).

Unfortunately I am not very familiar with the related literature, so I can't evaluate the work in that context.

---

### Official Review · Reviewer_RGGr · 2021-06-12
**A good paper with some flaws**

**Rating:** 7
**Confidence:** 3

**Review:**

The work presents an interesting solution approach to a relevant problem, namely an automated way of tuning learning rates during neural network training.
The approach to online tuning of the learning rate works by estimating the local loss landscape. This is achieved by perturbing the current learning rate value with a small epsilon and via multiple forward passes the loss landscape can be estimated and allows to determine if the learning rate should be increased or decreased and how strongly. By formulating a quadratic approximation to the loss, the optimal perturbation of the learning rate (according to the approximation) can be determined efficiently.
To avoid inducing to much computational overhead the learning rate is perturbed only every few minibatches and to avoid tuning only based on very local information the optimization is divided into a exploration end an exploitation phase.
Although the method and all its parts present a very interesting approach to adaptive learning rate schedules, the experiments fall a bit short. For example, only very simple learning rate schedules are compared to and the influence of elements such as the "rollback policy" are only briefly mentioned in text but not shown in an experiment. All in all I believe the work would benefit from a form of ablation study to show which parts of LRTuner contribute the most to the improved results reported over the basic hand-tuned schedules especially since the simple hand-tuned schedules adapt only every 30 epochs but the proposed method multiple times per epoch.
Further I believe it would very interesting to see a comparison to other Learning Rate Schedulers that were either manually designed but tuned via common HPO methods or potentially even learned learning rate schedules.
However the breadth of evaluated settings shows that the proposed heuristic for automatic learning rate scheduling is capable of adapting to various scenarios. I hope that the proposed future work can get rid of the LRT hyperparameters to have a fully automated learning rate scheduler without need to be tuned itself. All in all I think the paper is worthy of acceptance at the workshop and could facilitate interesting discussion between the participants.

Minor comments:
* Foot note 5 in text is at the beginning of a line. To connect it to "Adam optimizer" It should be sufficient to remove a space between "optimizer" and "\footnote".
* In the last line of the abstract: "... less optimization steps." -> "... fewer optimization steps."
* In Appendix E another Appendix is referenced but this reference is "??"
* The separation of Figure 6 into 3 parts seems rather confusing. For the accuracy and loss plots, the blue curve ends higher than the orange curve in the first column but then is lower in the second column. I think something is not correct in that plot. Figure 7 and 8 don't have such an artifact.

---

### Official Review · Reviewer_Joxm · 2021-06-19
**An explore-exploit method for tuning learning rates**

**Rating:** 6
**Confidence:** 3

**Review:**

The three main steps of the proposed methods are: 1) second-order Taylor expansion around a perturbation of the learning rate; 2) fitting the resulting quadratic function using few samples of the loss along the descent direction(rather than explicitly computing it); 3) Introduction of an exploration phase where only increase of learning rate are accepted.

I find these ideas simple and somewhat convincing. The addition of the explore phase seems quite important given the greediness of the method.

This being said, I would have appreciated a more thorough discussion of similarities and differences with other techniques (e.g. gradient-based techniques seem quite relevant, see e.g. [1] and [2] given the idea of small perturbations!) as well as proper ablation studies about these various choices (including the selection of the thresholding).
Without these, it's rather difficult to understand what's the impact of the various pieces of the method.
Probably the most striking absence is experiments with the method without the explore phase, in which only positive updates are accepted.

I must say that I also find Table 5 slightly discomforting, as it looks like the values of the hyperparameters used for each experiment are quite different and in a somewhat rather wide range. This is not a huge problem by itself, also because the authors do not put too much emphasis on the automation of the method, but must be taken into consideration when designing experiments to compare with other techniques and baselines.
The understand how much LRTuner is sensitive to its hyperparameters, you could consider running time-controlled experiments as in [2] placing a prior on the LRTuner hyperparameters (and those of other methods) and then running random search over these.


References
[1] Baydin, Atilim Gunes, et al. "Online learning rate adaptation with hypergradient descent." arXiv preprint arXiv:1703.04782 (2017).
[2] Donini, Michele, et al. "MARTHE: Scheduling the Learning Rate Via Online Hypergradients." arXiv preprint arXiv:1910.08525 (2019).

---

### Decision · Program_Chairs · 2021-06-21

Accept (Poster)